# Inhibition of SARS-CoV-2 Infection in Vero Cells by Bovine Lactoferrin under Different Iron-Saturation States

**DOI:** 10.3390/ph16101352

**Published:** 2023-09-25

**Authors:** Nathalia S. Alves, Adriana S. Azevedo, Brenda M. Dias, Ingrid S. Horbach, Bruno P. Setatino, Caio B. Denani, Waleska D. Schwarcz, Sheila Maria B. Lima, Sotiris Missailidis, Ana Paula D. Ano Bom, Andréa M. V. Silva, Débora F. Barreto Vieira, Marcos Alexandre N. Silva, Caroline A. Barros, Carlos Alberto M. Carvalho, Rafael B. Gonçalves

**Affiliations:** 1Instituto de Tecnologia em Imunobiológicos, Fundação Oswaldo Cruz, Rio de Janeiro 21040-900, RJ, Brazil; nathalia.alves@bio.fiocruz.br (N.S.A.); adriana.soares@bio.fiocruz.br (A.S.A.); brenda.dias@bio.fiocruz.br (B.M.D.); ingrid.horbach@bio.fiocruz.br (I.S.H.); bruno.setatino@bio.fiocruz.br (B.P.S.); caio.denani@bio.fiocruz.br (C.B.D.); waleska.dias@bio.fiocruz.br (W.D.S.); smaria@bio.fiocruz.br (S.M.B.L.); sotiris.missailidis@bio.fiocruz.br (S.M.); adinis@bio.fiocruz.br (A.P.D.A.B.); amarques@bio.fiocruz.br (A.M.V.S.); 2Laboratório de Morfologia e Morfogênese Viral, Instituto Oswaldo Cruz, Fundação Oswaldo Cruz, Rio de Janeiro 21040-900, RJ, Brazil; barreto@ioc.fiocruz.br (D.F.B.V.); marquinhosans@gmail.com (M.A.N.S.); 3Instituto de Bioquímica Médica Leopoldo de Meis, Universidade Federal do Rio de Janeiro, Rio de Janeiro 21941-902, RJ, Brazil; carol.augustobarros@gmail.com; 4Instituto Federal de Educação, Ciência e Tecnologia do Rio de Janeiro, Rio de Janeiro 20270-021, RJ, Brazil; 5Departamento de Patologia, Centro de Ciências Biológicas e da Saúde, Universidade do Estado do Pará, Belém 66095-662, PA, Brazil; 6Departamento de Bioquímica, Instituto Biomédico, Universidade Federal do Estado do Rio de Janeiro, Rio de Janeiro 20211-040, RJ, Brazil; rafael.braga@unirio.br

**Keywords:** antiviral activity, lactoferrin, SARS-CoV-2

## Abstract

Despite the rapid mass vaccination against COVID-19, the emergence of new SARS-CoV-2 variants of concern, such as omicron, is still a great distress, and new therapeutic options are needed. Bovine lactoferrin (bLf), a multifunctional iron-binding glycoprotein available in unsaturated (apo-bLf) and saturated (holo-bLf) forms, has been shown to exert broad-spectrum antiviral activity against many viruses. In this study, we evaluated the efficacy of both forms of bLf at 1 mg/mL against infection of Vero cells by SARS-CoV-2. As assessed with antiviral assays, an equivalent significant reduction in virus infection by about 70% was observed when either form of bLf was present throughout the infection procedure with the SARS-CoV-2 ancestral or omicron strain. This inhibitory effect seemed to be concentrated during the early steps of virus infection, since a significant reduction in its efficiency by about 60% was observed when apo- or holo-bLf were incubated with the cells before or during virus addition, with no significant difference between the antiviral effects of the distinct iron-saturation states of the protein. However, an ultrastructural analysis of bLf treatment during the early steps of virus infection revealed that holo-bLf was somewhat more effective than apo-bLf in inhibiting virus entry. Together, these data suggest that bLf mainly acts in the early events of SARS-CoV-2 infection and is effective against the ancestral virus as well as its omicron variant. Considering that there are no effective treatments to COVID-19 with tolerable toxicity yet, bLf shows up as a promising candidate.

## 1. Introduction

Coronaviruses (CoVs) belong to the family *Coronaviridae* of the order *Nidovirales* [1,2] and cause various diseases in animals and humans, mainly enteritis and respiratory infections. The zoonotic outbreak of severe acute respiratory syndrome (SARS) that occurred in 2002 and Middle East respiratory syndrome (MERS) in 2012 showed their pathogenic potential, being responsible for hundreds of deaths [3]. At the end of 2019, a pandemic began with the coronavirus disease 2019 (COVID-19), caused by a new species of CoV, the Severe Acute Respiratory Syndrome Coronavirus 2 (SARS-CoV-2), leading to millions of deaths worldwide [4].

The main structure of CoVs is composed of four distinct proteins known as spike (S), envelope (E), membrane (M) and nucleocapsid (N) proteins [1,2]. The CoV genome is made up of a monopartite single-stranded ribonucleic acid (ssRNA) with positive sense [5]. Virions are spherical and have spiky projections on their surface that give them a corona feature, recognized via electron microscopy. The nucleocapsid is located inside the envelope and has a helical symmetry [6]. Binding of SARS-CoV-2 to the host cell occurs through the interaction of the S protein with angiotensin-converting enzyme 2 (ACE2). The internalization process is mediated by transmembrane serine protease 2 (TMPRSS2) and cathepsin L (CTSL), host enzymes mainly expressed in the plasma membrane and endolysosomes, respectively, of lung, gastrointestinal tract and kidney cells that prime the S protein for membrane fusion [7]. The greater tropism of the virus for cells of the lung alveolar epithelium, kidneys and small intestine is also related to the greater expression of ACE2 in these tissues [8].

After the emergence of alpha, beta, gamma and delta SARS-CoV-2 variants of concern (VoCs) over several countries from different continents, a new SARS-CoV-2 VoC—so-called omicron—was discovered in Botswana, Africa, and showed an even higher rate of infectivity and immune escape, spreading rapidly across the world [9]. This high infectivity is due to over 30 amino acid substitutions in the S protein that inhibit the neutralization of the viral particle by host antibodies. In addition, five S protein mutations (G339D, N440K, S477N, T478K and N501Y) are related to increased interaction of the viral particle with the human ACE2 receptor [10]. The omicron variant is about 2.8 and 10 times more infectious than the delta variant and the ancestral virus, respectively [11], and the appearance of omicron subvariants with additional mutations further contributed to the wide spread of SARS-CoV-2 around the world [12]. In the current pandemic context, the development of new strategies to inhibit SARS-CoV-2 infection is highly necessary.

Lactoferrin (Lf) is an iron-binding glycoprotein present in various biological fluids such as milk, saliva and blood [13]. Lf has a great affinity for iron ions—especially Fe^3+^—and can be found in unsaturated (apo-Lf) and saturated (holo-Lf) forms depending on its binding to these ions [14]. The presence of iron ions in the structure of Lf promotes changes in its tertiary conformation, which seems to increase its stability under denaturing chemical and physical conditions [15].

The multifunctional character of Lf has been widely investigated, more so using its bovine version (bLf) than its human one (hLf), revealing antimicrobial, antifungal, antitumor, immunomodulatory and antiviral activities [16,17,18]. Over the years, the antiviral activity of bLf has been tested with several viruses such as Mayaro, Chikungunya, Zika, Dengue and Herpes Simplex viruses, as well as adenoviruses and rhinoviruses [19,20,21,22,23,24]. Recently, our group has pioneered the demonstration that bLf reduces progeny SARS-CoV-2 yield by up to ~84.6% in African green monkey kidney epithelial (Vero E6) cells and ~68.6% in adenocarcinomic human alveolar basal epithelial (A549) cells at 1 mg/mL [25], a concentration previously shown to have no cytotoxicity for both apo- and holo-bLf [15]. Such an inhibitory effect of bLf on SARS-CoV-2 was further confirmed by several other groups [26,27,28,29].

Although apo-bLf was already shown to be more potent than holo-bLf against some virus species [18], none of these previous works evaluated the effect of iron saturation on the anticoronaviral activity of the protein. Thus, our aim in the current study was to comparatively investigate the antiviral activity of apo- and holo-bLf at 1 mg/mL against infection of Vero cells by SARS-CoV-2.

## 2. Results and Discussion

### 2.1. Overall Antiviral Effect of Apo- or Holo-bLf on SARS-CoV-2 Infection

Comparisons between the sensitivities of infection by SARS-CoV-2 ancestral and omicron strains to bLf under different iron-saturation states were carried out by incubating Vero cell monolayers with 1 mg/mL apo- or holo-bLf during all stages of the infection procedure (Figure 1). There was a large reduction in the percentage of infection with either of the protein forms for SARS-CoV-2 ancestral (~71% and ~74% for apo- and holo-bLf, respectively) as well as omicron (~65% and ~67% for apo- and holo-bLf, respectively) strains, although the observed differences between apo- and holo-bLf were not significant. Also, no significant differences were observed between the percentages of infection when comparing both virus strains.

This finding is in line with a recent study showing that 1 mg/mL apo- or holo-hLf derived from the milk of transgenic goats promoted a similar pronounced reduction in the titers of SARS-CoV-2 ancestral and delta strains during infection of Vero E6 cells [30]. Since our data revealed that SARS-CoV-2 ancestral and omicron strains were also equivalent in terms of sensitivity to apo- or holo-bLf during infection, the ancestral one was selected for the next experiments as a precautionary measure due to its lower risk of community spread.

### 2.2. Antiviral Effect of Apo- or Holo-bLf on Specific Stages of SARS-CoV-2 Infection

To assess at which stage of SARS-CoV-2 infection bLf acts, Vero cell monolayers were incubated with 1 mg/mL apo- or holo-bLf before, during or after the infection procedure, or untreated as a control (Figure 2). There was a great reduction in the percentage of virus infection when either of the protein forms were added before (~55% and ~70% for apo- and holo-bLf, respectively) or during (~57% and ~60% for apo- and holo-bLf, respectively) the infection procedure. However, no significant reduction in the percentage of virus infection was observed when either of the protein forms were added after the infection procedure. There were also no significant differences between the percentages of virus infection when comparing the effects of apo- and holo-bLf.

This result agrees with previous findings reporting that bLf was able to inhibit the replication of a pseudotyped SARS-CoV-2 (used as a surrogate for the infectious SARS-CoV-2) in human rhabdomyosarcoma (RD) cells at the attachment stage, but not viral entry and post-entry stages, even though the authors have not considered the iron-saturation state of bLf in their assays [31]. However, this does not seem to be the only stage targeted by bLf in respiratory virus infections, as rhinovirus B-14 entry and post-entry stages in human cervical adenocarcinoma (H1-HeLa) cells were already shown to also be affected by bLf [24].

### 2.3. Inhibition of SARS-CoV-2 Entry by Apo- or Holo-bLf

Ultrastructural analysis of SARS-CoV-2-infected Vero cells treated or not with 1 mg/mL apo- or holo-bLf revealed increased plasma membrane projections, proliferation of intracellular vesicles, rough endoplasmic reticulum thickness, number of highly electron-dense ribosomes and vacuolization in relation to mock-infected cells under the same treatment conditions (Figure 3). As expected, no virus-resembling structures were observed in mock-infected cells, regardless of whether they were treated or not with bLf. On the other hand, SARS-CoV-2-infected cells showed ~4.43 viruses/cell for the control condition, ~0.57 viruses/cell for the apo-bLf condition and ~0.00 viruses/cell for the holo-bLf condition.

One of the ways that bLf hinders virus infection is by preventing its internalization in the host cell by blocking surface heparan sulfate proteoglycans (HSPGs), which are used by many viruses as a non-specific initial adhesion molecule [32]. Indeed, it has already been reported that bLf prevents the internalization of SARS-CoV and SARS-CoV-2 by interacting with HSPGs [31,33]. Furthermore, other works have demonstrated the interaction of bLf with the host cell ACE2 and the SARS-CoV-2 S protein [26,32]. Regarding potential mechanistic differences between the anticoronaviral activity of bLf, it is important to highlight that a previous study from our group has shown that holo-bLf is internalized faster than apo-bLf in Vero cells, which may reflect distinct affinities of iron-free and iron-saturated forms of the protein to its specific receptor (LfR) present in this cell lineage [15].

## 3. Materials and Methods

### 3.1. Virus Source and Cell Culture

SARS-CoV-2 strains (ancestral Wuhan strain—GISAID EPI_ISL_414045—and omicron BA.1 strain—GISAID EPI_ISL_8430488) were kindly provided by the Laboratory of Respiratory Viruses and Measles at IOC/Fiocruz (Rio de Janeiro, RJ, Brazil). Vero cells (CCL-81—American Type Culture Collection, Gaithersburg, MD, USA) were cultured as monolayers at 37 °C in a humidified atmosphere with 5% CO_2_ in 199 medium with Earle’s salts (E199) (Sigma-Aldrich, Saint Louis, MO, USA) supplemented with 5% fetal bovine serum (Invitrogen, Waltham, MA, USA), 1% gentamicin sulfate (Santisa Laboratório Farmacêutico, Bauru, SP, Brazil) and 0.4% amphotericin B (Gibco, Grand Island, NY, USA).

### 3.2. Preparation of Apo- and Holo-bLf

Apo-bLf dry powder (CAS BioSciences, New York, NY, USA) was dissolved to a concentration of 100 mg/mL in phosphate-buffered saline (PBS) and centrifuged at 6000 rpm for 5 min at 4 °C to remove the cellulose excipient. Holo-bLf was obtained from apo-bLf (10 mg/mL) via its dilution in 10 mM Tris and 75 mM NaCl (pH 7.2) followed by the addition of an FeNTA solution—9.9 mM Fe(NO_3_)_3_ and 8.5 mM nitrilotriacetic acid (pH 7.0)—in a 2:1 proportion. To remove the excess of iron ions from the solution, the sample was dialyzed against the standard buffer described above for 48 h at 4 °C. Both apo- and holo-bLf were filtered through a 0.22 µm syringe-driven filter unit (Millipore, Billerica, MA, USA), aliquoted and stored at 4 °C until further use. The iron load of the protein was determined on the spectrophotometer Multiskan GO (Thermo Scientific, Waltham, MA, USA) at 465 nm in standard buffer.

### 3.3. Antiviral Assays

The overall antiviral effect of apo- or holo-bLf against SARS-CoV-2 was assessed by incubating Vero cell monolayers in 24-well plates at ~80% confluence with 1 mg/mL bLf throughout the infection procedure (i.e., before, during and after virus addition). A time-of-addition assay was also performed by varying the stage of infection in which apo- or holo-bLf was added: briefly, Vero cell monolayers in 24-well plates were incubated with 1 mg/mL bLf before, during or after the infection procedure, or left untreated as a control. Infection was performed with 60–100 plaque-forming units (PFUs)/well of SARS-CoV-2, and all steps occurred at 37 °C and 5% CO_2_. After incubation with the virus, the E199 medium was discarded and cells were overlaid with a semi-solid version of the medium containing 1.5% or 1.25% carboxymethylcellulose (CMC), depending on the SARS-CoV-2 strain used (ancestral and omicron, respectively). At the end of the assay, cells were fixed with a 1.25% formalin solution and stained with crystal violet to allow for plaque counting via visual inspection.

### 3.4. Transmission Electron Microscopy

For transmission electron microscopy analysis, Vero cell monolayers were incubated with SARS-CoV-2 under a multiplicity of infection of 5 PFU/cell in the presence or absence of 1 mg/mL apo-bLf or holo-bLf for 30 min at 4 °C to allow for virus binding and then incubated for an additional 30 min at 37 °C to allow for virus entry. Thereafter, cells were dissociated via trypsinization, fixed with 1% glutaraldehyde in 0.1 M sodium cacodylate buffer (pH 7.2), post-fixed in 1% buffered osmium tetroxide, dehydrated in acetone, embedded in epoxy resin and polymerized at 60 °C for 72 h. Ultrathin sections (50–70 nm) were obtained from the resin blocks, picked up using copper grids (300 mesh) and observed on a HT7800 transmission electron microscope (Hitachi, Chiyoda, TK, Japan). Mock-infected cells subjected to the bLf treatment and SARS-CoV-2-infected cells not subjected to the bLf treatment were used as controls.

### 3.5. Statistical Analyses

Gaussian distribution was confirmed using Kolmogorov–Smirnov normality test with Dallal–Wilkinson–Lilliefor *p* value, and then statistical hypothesis testing was performed using ordinary two-way ANOVA, followed by post-hoc Tukey’s multiple comparison test on Prism 9 (GraphPad, San Diego, CA, USA). *p* values less than or equal to 0.05 were considered as indicative of statistically significant differences.

## 4. Conclusions

Our results demonstrated that both apo- and holo-bLf at 1 mg/mL are effective against in vitro infection of Vero cells by SARS-CoV-2 ancestral and omicron strains, inhibiting the early steps of virus infection with no significant differences in the effect size. Further studies are still required to address whether the molecular mechanisms related to such an inhibitory activity differ according to the iron-saturation state of the protein.

## Figures and Tables

**Figure 1 pharmaceuticals-16-01352-f001:**
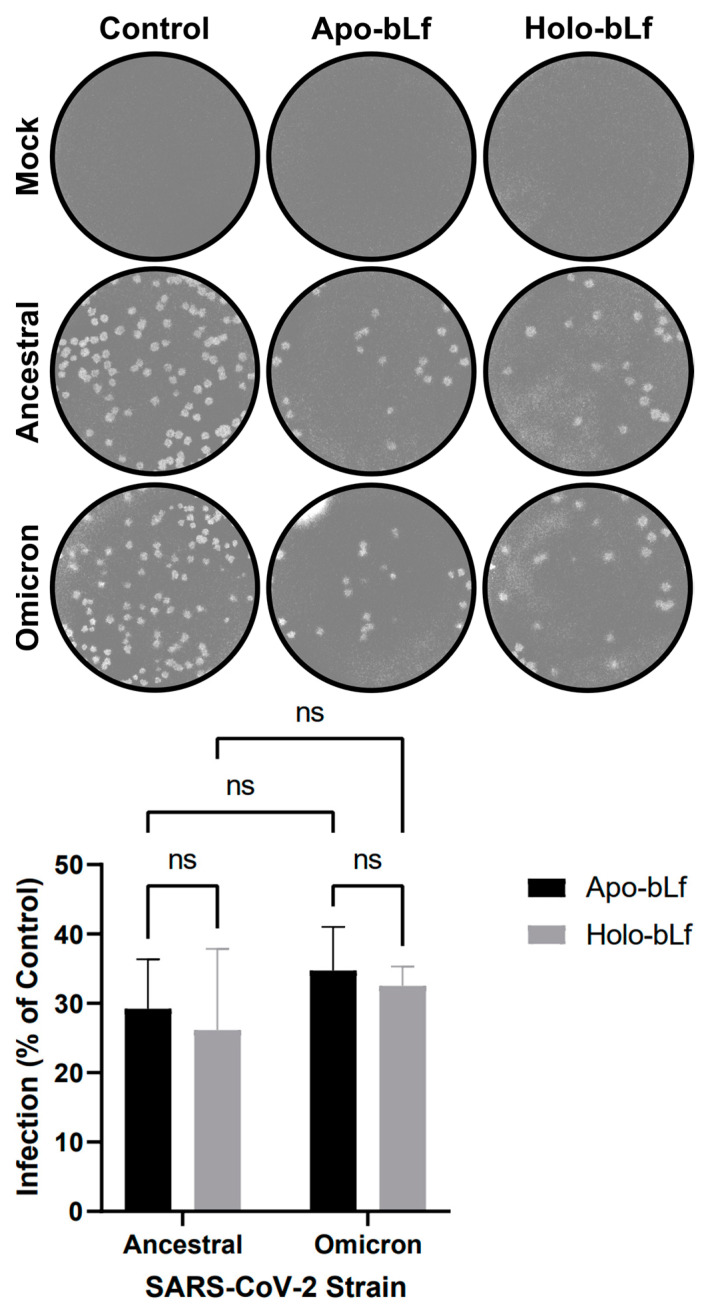
Overall antiviral effect of apo- or holo-bLf on SARS-CoV-2 infection. Vero cells were treated with 1 mg/mL apo- or holo-bLf throughout the infection procedure (i.e., before, during and after virus addition). At 72 h post-infection, cells were stained (**top**) and plaques were counted (**bottom**) to determine the efficiency of infection in relation to untreated cells (control). Data were obtained from three independent experiments in triplicates and plotted as mean ± standard deviation. No statistically significant differences between the groups were detected (ns: *p* > 0.05).

**Figure 2 pharmaceuticals-16-01352-f002:**
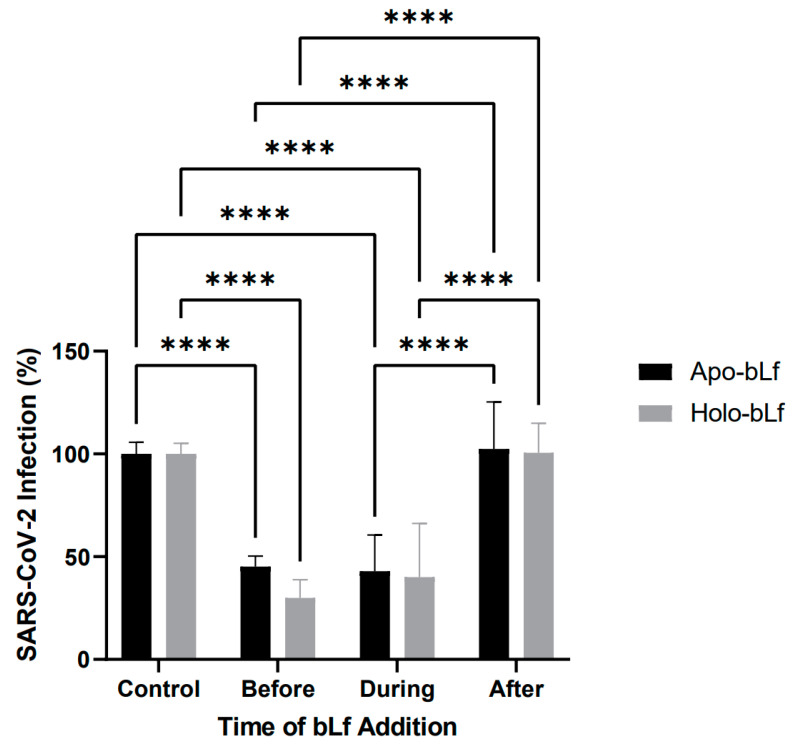
Antiviral effect of apo- or holo-bLf on specific stages of SARS-CoV-2 infection. A time-of-addition assay was carried out in Vero cells either untreated (control) or treated with 1 mg/mL apo- or holo-bLf for 1 h before, for 1 h during or for 72 h after the infection procedure. At 72 h post-infection, cells were stained and plaques were counted to determine the efficiency of infection. Data were obtained from three independent experiments in triplicates and plotted as mean ± standard deviation. Only statistically significant differences within the same group were plotted (****: *p* ≤ 0.0001).

**Figure 3 pharmaceuticals-16-01352-f003:**
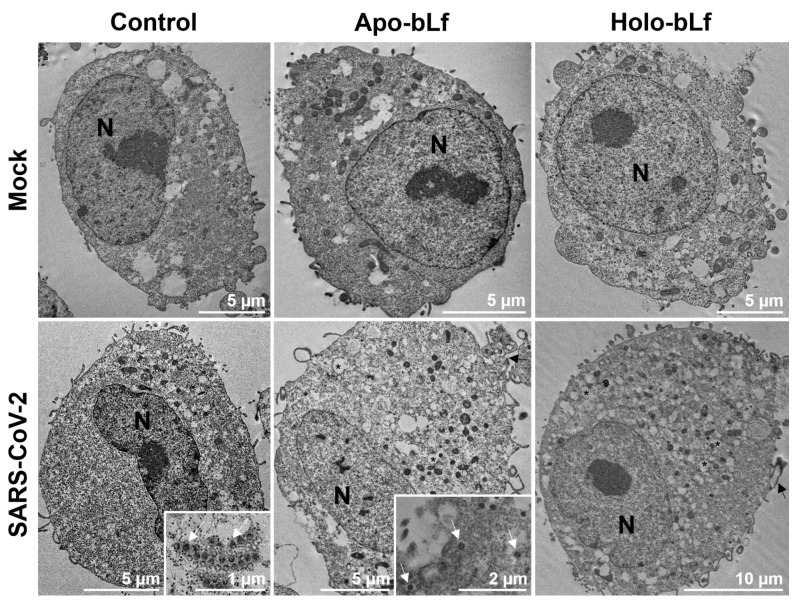
Inhibition of SARS-CoV-2 entry by apo- or holo-bLf. An ultrastructural analysis was performed in Vero cells either untreated (control) or treated with 1 mg/mL apo- or holo-bLf for 30 min during virus adsorption at 4 °C. After incubation for an additional 30 min at 37 °C, cells were dissociated and processed for transmission electron microscopy. SARS-CoV-2-infected cells were compared to mock-infected cells to assess the effect of virus entry on cell morphology. Images are representative of at least 14 cells for each condition (N: nucleus; *: intracellular vesicle; black arrow: plasma membrane projection; white arrow: SARS-CoV-2 particle).

## Data Availability

Data is contained within the article.

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
