# Peer review of "Inhibition of SARS-CoV-2 Infection in Vero Cells by Bovine Lactoferrin under Different Iron-Saturation States"

_pharmaceuticals, 2023, doi:10.3390/ph16101352_

Round 1
Reviewer 1 Report
The authors comparatively investigate the antiviral activity of apo- and holo-bLf against infection of Vero 93 cells by ancestral and omicron strains of SARS-CoV-2. The questions are:
1. Title: as the work was made in Vero cells, the authors can change the title to do it more specific.
2. Although the results are good, it can be reinforced that this is a comparative study between apo- and holo-LF. On the other hand, it would be better to be continued the experiments with the omicron strain.
3. What was the LF concentration in the apo- and holo-LF?
Author Response
We are grateful to Reviewer 1 for constructive criticism and for the time taken to review our manuscript, which helped to improve its quality. We have provided a point-by-point response to each of the comments and suggestions on our study below, indicating precisely where the changes were made in the text and/or its related figures:
Point 1: Title: as the work was made in Vero cells, the authors can change the title to do it more specific.
Response 1: Thank you for this observation; as suggested, the title has been changed to “Inhibition of SARS-CoV-2 infection in Vero cells by bovine lactoferrin under different iron-saturation states” in the revised manuscript (lines 2-3).
Point 2: Although the results are good, it can be reinforced that this is a comparative study between apo- and holo-LF. On the other hand, it would be better to be continued the experiments with the omicron strain.
Response 2: Indeed, the main aim of this study was to comparatively investigate the antiviral activity of apo- and holo-bLf against infection of Vero cells by SARS-CoV-2. To avoid misunderstandings, mentions to ancestral and omicron strains of SARS-CoV-2 have been removed from aim statements in the abstract (lines 26-27) and introduction (lines 94-96) sections of the revised manuscript. As ancestral and omicron strains were equivalent in terms of sensitivity to apo- or holo-bLf during infection, excluding the latter SARS-CoV-2 strain from the rest of the experiments was a precautionary measure due to its higher risk of community spread. This explanation has been added to lines 119-121 of the revised manuscript.
Point 3: What was the LF concentration in the apo- and holo-LF?
Response 3: Lf concentration was 1 mg/mL in all experiments of this study regardless of its iron-saturation state, as indicated in the subsections 3.3 and 3.4 as well as in the legends of Figures 1, 2 and 3. To make it even more explicit, this information has been also included in the abstract (lines 26-27), introduction (line 95) and conclusions (line 231) sections of the revised manuscript.
Author Response
We are grateful to Reviewer 2 for constructive criticism and for the time taken to review our manuscript, which helped to improve its quality. We have provided a point-by-point response to each of the comments and suggestions on our study below, indicating precisely where the changes were made in the text and/or its related figures: Point 1: Line 37 – Dose the author believe the production of bLf is cost effectives? Has the author provided a acceptable document to prove it in the paper? Response 1: Although bLf is naturally found in cow's milk, its relatively low concentration and current resource-intensive extraction processes may limit access. However, according to a recently published market research report, the global bLf market size is projected to grow from USD 666.6 million in 2022 to USD 1,850.3 million by 2029 at a compound annual growth rate (CAGR) of 15.7% in the forecast period, favoring the cost effectiveness of its production (https://www.fortunebusinessinsights.com/industry-reports/bovine-lactoferrin-market-101656). Morever, the world’s first animal-free bLf has been recently announced by a producer of functional dairy proteins through the use of precision fermentation, unlocking an affordable source of bLf (https://www.futureofproteinproduction.com/post/first-ever-animal-free-lactoferrin-in-world-first-tasting). Anyway, the mention of low cost for COVID-19 treatments was removed from the abstract section of the revised manuscipt (line 37) to avoid a possible misinterpretation. Point 2: Line 57 – Please indicate the source of this protein. How this protein express? Response 2: TMPRSS2 is a host cell protease mainly expressed in the plasma membrane of lung, gastrointestinal tract and kidney cells, where it can prime the virus S protein for membrane fusion. When TMPRSS2 is absent, the endolysosomal protease cathepsin L (CTSL) – also highly expressed in the above tissues – plays this critical role during SARS-CoV-2 internalization. This information has been included in lines 56-60 of the revised manuscript. Point 3: Line 99 – Please explain the reasons of selection this concentration of bLf (1 mg/mL) Response 3: As explained in lines 85-88 of the revised manuscript, a pilot study of our group has shown that the bLf concentration of 1 mg/mL was the most effective in reducing progeny virus yield during SARS-CoV-2 infection (Carvalho et al., 2020 – bioRxiv: 2020.05.13.093781). Point 4: Line 146 – Did the author test the effect of both Apo- or Holo-bLf (1 mg/mL) on the viability and morphological properties of Vero cells? Response 4: Yes, a previous work from our group has shown that bLf is non-cytotoxic at 1 mg/mL in Vero cells, regardless of its iron-saturation state (Barros et al., 2021 – Heliyon 7: e08087). This information can be found in line 89 of the revised manuscript. Point 5: Line 182 – Please indicate the form of Apo-bLf (Powder?) Response 5: Apo-bLf was purchased as a dry powder formulation and dissolved in PBS to reach the desired stock concentration. This information has been added to line 187 of the revised manuscript. Point 6: Line 190 – Please change this sentences as follow: Both apo- and holo-bLf were filtered through 0.22-mm syringe-driven filter unit (Millipore, Billerica, Ma, USA) aliquoted and stored at 4 C until further use Response 6: Thank you for the recommendation; this sentence has been changed in the revised manuscript (lines 193-195) as indicated. Point 7: Line 220 – Please indicate the number of replicate for each treatment. Response 7: Thank you for the observation; as requested, the number of replicates for each independent experiment has been indicate in the legends of Figures 1 (line 113) and 2 (line 137) of the revised manuscript, where it now reads: “Data were obtained from three independent experiments in triplicates”.Reviewer 3 Report
Thank you for the opportunity to review the paper.
Comments and suggestions:
Line 99 – Please mention “Vero Cells”. Because, the results section precedes methods and materials, it will be helpful to add sentence on the type of cells (e.g., Vero)
Figure 1. Please consider adding images.
Line 120 – Please add information on cell culture parameters, including but not limited to cell culture media composition, incubator conditions.
Line 180 – Please add antibiotics information
Line 184 - Please correct the pore size of the syringe driven filter unit
Line 219 - Please consider rewriting the sentence and emphasizing that the normality test precedes the two-way ANOVA, which is followed by post-hoc Tukey’s multiple comparison test
Others - Please consider rewording the title to emphasis the experiments were performed using kidney cells, from African green monkey
The results are interesting and the work would be enhanced, if a second cell line was also used
Quality of English language - Minor editing
Author Response
We are grateful to Reviewer 3 for constructive criticism and for the time taken to review our manuscript, which helped to improve its quality. We have provided a point-by-point response to each of the comments and suggestions on our study below, indicating precisely where the changes were made in the text and/or its related figures:
Point 1: Line 99 – Please mention “Vero Cells”. Because, the results section precedes methods and materials, it will be helpful to add sentence on the type of cells (e.g., Vero)
Response 1: Thank you for this observation; as suggested, the cell line used has been indicated in the beginning of subsections 2.1 (line 101), 2.2 (line 123) and 2.3 (line 149) of the revised manuscript.
Point 2: Figure 1. Please consider adding images.
Response 2: In the revised manuscript, we have included images of representative plates from the first antiviral assay as a top panel in Figure 1, revealing that apo- and holo-bLf promote a reduction in the number of SARS-CoV-2 plaques without apparently affecting their size.
Point 3: Line 120 – Please add information on cell culture parameters, including but not limited to cell culture media composition, incubator conditions.
Response 3: Detailed information on cell culture parameters – including cell culture medium composition and incubation conditions – had already been provided in subsections 3.1 and 3.3 of the original manuscript, but they have been incremented in the revised manuscript to show additional information (lines 180-185 and 199-206).
Point 4: Line 180 – Please add antibiotics information
Response 4: As requested, information on the antibiotics used has been added to lines 184-185 of the revised manuscript, where it now reads: “1% gentamicin sulfate (Santisa Laboratório Farmacêutico, Bauru, SP, Brazil) and 0.4% amphotericin B (Gibco, Grand Island, NY, USA)”.
Point 5: Line 184 - Please correct the pore size of the syringe driven filter unit
Response 5: Thank you for this observation; the pore size of the syringe-driven filter unit has been correct from 0.22 mm to 0.22 µm in the revised manuscript (line 194).
Point 6: Line 219 - Please consider rewriting the sentence and emphasizing that the normality test precedes the two-way ANOVA, which is followed by post-hoc Tukey’s multiple comparison test
Response 6: The sentence on statistical analyses has been rewrited as recommended in lines 225-228 of the revised manuscript, where it now reads: “Gaussian distribution was confirmed by Kolmogorov-Smirnov normality test with Dallal-Wilkinson-Lilliefor P value and then statistical hypothesis testing was performed using ordinary two-way ANOVA followed by post-hoc Tukey’s multiple comparison test on Prism 9 (GraphPad, San Diego, CA, USA)”.
Point 7: Others - Please consider rewording the title to emphasis the experiments were performed using kidney cells, from African green monkey
Response 7: Thank you for this observation; to make the title even more specific, it has been changed to “Inhibition of SARS-CoV-2 infection in Vero cells by bovine lactoferrin under different iron-saturation states” in the revised manuscript (lines 2-3).
Point 8: The results are interesting and the work would be enhanced, if a second cell line was also used
Response 8: We appreciate the feedback on our results. Although a second cell line might provide additional information, repeating the antiviral assays with another model in the context of a biosafety level 3 (BSL-3) facility would far exceed the journal’s strict deadline for responding to reviewers. However, in vitro studies on SARS-CoV-2 – as well as on bLf – have been extensively carried out in Vero cells, whose selection thus allows drawing more direct comparisons between data from different laboratories. Indeed, this cell line has become one of the most common mammalian immortalized cell lines used in research, gaining prominence in the field of virology for being susceptible to a wide range of viruses (Ammerman et al., 2008 – Curr. Protoc. Microbiol. 4: 1-10).
Round 2
Reviewer 1 Report
The manuscript was corrected appropriately.